# Twenty Years of Research in Seabass and Seabream Welfare during Slaughter

**DOI:** 10.3390/ani11082164

**Published:** 2021-07-22

**Authors:** Ignacio de la Rosa, Pedro L. Castro, Rafael Ginés

**Affiliations:** 1Departamento de Ciencias Agroforestales, Universidad de Huelva, 21004 Huelva, Spain; ignacio.delarosa@dcaf.uhu.es; 2Grupo de Investigación en Acuicultura-GIA, IU-ECOAQUA, Universidad de Las Palmas de Gran Canaria, 35214 Telde, Spain; rafael.gines@ulpgc.es

**Keywords:** seabass, seabream, welfare, stunning, slaughtering

## Abstract

**Simple Summary:**

Sea cage farms dominate European aquaculture production of seabass (*Dicentrarchus labrax*) and gilthead seabream (*Sparus aurata*). It means that to complete the commercialization process, fish must be crowded in a net, lifted from the rearing cage, and placed in a stunning/slaughtering tank during the extraction procedure. Brailing and pumping are the two techniques used. The brailing involves the use of a large net that is hoisted by a crane, and the fish and water are released from the brail by opening the closed end of the net with a release. The fish enter water through a pipe and pass through a grid that removes the water before being placed in the stunning/slaughtering tank. This paper examines the scientific progress made in these areas over the last two decades in relation to farmed seabass and seabream describing the consequences of different methodologies on the time fish takes to reach the unconscious stage, the different concentrations of stress indicators in plasma, and the evolution of flesh quality related to spoilage during fish shelf-life.

**Abstract:**

The behavioural responses of fish to a stressful situation must be considered an adverse reaction caused by the perception of pain. Consequently, the handling prior to stunning and the immediacy of loss consciousness following stunning are the aspects to take into account during the slaughtering process. The most common commercial stunning method in seabream and seabass is based on hypothermia, but other methods such as electrical stunning, carbon dioxide narcosis or anaesthetic with clove oil, are discussed in relation to the time to reach the unconsciousness stage and some welfare indicators. Although seawater plus ice slurry is currently accepted in some guidelines of fish welfare well practices at slaughter, it cannot be considered completely adequate due to the deferred speed at which cause loss of consciousness. New methods of incorporating some kind of anaesthetic in the stunning tank could be a solution to minimize the impact on the welfare of seabass and seabream at slaughtering.

## 1. Introduction

Animal welfare evaluation should be promoted so that decisions are made based on scientific evidence rather than emotion, with the understanding that the concept of welfare is a characteristic of an animal, not something given to it, and can be precisely measured [1]. In the case of fish production, the objective measurement of animal welfare is an issue that must be addressed in order to promote adequate guidelines for the levels of acceptability management [2], particularly during the stunning and slaughter processes. The most contentious research area in non-mammalian welfare is the debate over whether fish feel pain [3], but given that fish exhibit behavioural and physiological responses similar to those found in mammals, there is a large scientific consensus that there is no adequate basis for denying them conscious pain experiences [4].

If pain is defined as an unpleasant sensory and emotional experience associated with actual or potential harm, an animal must have sentience in order to experience pain. According to Chandroo et al. [5], fish suffer in ways similar to tetrapods because anatomical, pharmacological, and behavioural data indicate that affective states such as pain, fear, and stress are likely to be experienced. Nevertheless, arguments against the fact that fish feel pain repeatedly appear over the capacity for non-mammalian species to experience the discomfort or suffering rather than a nociceptive reflex. Pain is caused by neural processing in the brain that necessitates structural connectivity and the presence of a cortex, which fish lack [6]. Fish may not have the complex brains of the higher mammals, but they do have a nervous system that can detect noxious stimulation. Such an experience does not need a cortex because the experience is raw, tied directly to the immediate damage, and is an objective extension of that damage, which drives the aversive behavioural responses [7,8]. In fact, fish have nociceptors that detect noxious stimuli and brain pathways that process nociception signals in the same way that vertebrates do [9], so their behaviour responses after noxious stimuli administration are not simply reflexes but rather indicators of pain perception [10].

The possibility that fish are sentient and, as a result, experience pain and suffering has become a major topic in aquaculture in order to provide appropriate conditions during slaughter, i.e., to be unconscious and insensible when slaughtered [11]. Simple risk analysis on a simple neural system shows that the probability that fish can feel pain is not negligible [12], and a precautionary principle for welfare consideration should still advance animal welfare protection [13]. As a result, the best practice would be to provide fish with the same level of protection that any other vertebrate receives [14]. Despite the fact that fishes are very different from us and are unlikely to have a capacity for awareness of pain or emotional feelings that meaningfully resemble our own [15], a strong alternative view is that complex animals with sophisticated behaviour probably have the capacity for suffering, though it may differ in degree and kind from the human experience [16]. In any case, sentient animals in our care must be kept in comfortable conditions that maximize their health and welfare, and they must be slaughtered as quickly and painlessly as possible [11]. The aquaculture industry should be governed by ethical principles ensuring the health and welfare of fish, including humane slaughter [12], which should incorporate them into a holistic assessment for fish management, not as a purely scientific analysis, assess available alternatives and take into account new knowledge to recalculate ethical stress in the new perspectives [17].

Taking into consideration that the total aquaculture production of seabass and seabream increased from just under 8 thousand tons in 1990 to 522 thousand tons in 2019 [18], it seems pertinent to discuss the scientific evidence on slaughter methodologies to try to guarantee the best welfare conditions for both species.

## 2. Impact of Stunning on Farmed Fish Welfare

The goal of optimal fish slaughter is to eliminate needless stress and agony during the procedure [19]. Humane slaughter methods are designed to bring about the rapid loss of consciousness and, ultimately, a complete loss of brain function in animals destined for use as food. This means minimizing or eliminating anxiety, pain, and distress associated with terminating the lives of the fish [20]. In addition, the approach utilized to kill the fish should do so fast after successful stunning to avoid regaining consciousness [2].

Clearly, the development of appropriate commercial technologies to promote a humane slaughter for farmed fish is an active area of research, linked to a growing awareness in the aquaculture industry about the importance of ensuring that stunning takes place under the best conditions to cause loss of consciousness until the fish dies. Thus, it is necessary to identify stressful situations early enough so that an intervention can take place before detrimental effects occur [21] and determine how quickly the fish is rendered insensible, which is difficult in practice [22]. Finally, the induction of unconsciousness should not cause suffering even if the methodology used does not result in an immediate loss of consciousness [23].

It is understood that consumer acceptance of aquaculture products will be increasingly influenced by the extent to which the industry is perceived to be dealing with fish welfare, obviously including the time of slaughter [24], with a growing insistence among consumers that the animals they eat were well treated [11]. The challenge for the fish farming industry in this context is to demonstrate that this activity is conducted in an ethical and humane manner [25].

### 2.1. Pre-Slaughter Handling

The application of slaughter technology varies by species, but it is well-established in the majority of segments of the food fish industry to achieve product quality control, efficiency, and processor safety [26]. The slaughtering process in a fish farm, in this case rearing the fish in sea cages, consists of a starvation period to empty the gut, crowding and collecting or pumping to remove the fish from the water, stunning, and killing. Despite the fact that most research focuses on the stress experienced during the slaughtering process, the negative prior handling is frequently overlooked [17]. As a result, rough handling during crowding and repeated catching cause additional stress, resulting in increased cortisol and haematocrit levels [27]. The impact of entire processes on welfare varies significantly depending on the species [28], and it also affects fish quality because pre-slaughter handling and slaughter methods start an irreversible process of flesh degradation [29]. Indeed, a multidisciplinary approach that considers animal behaviour as well as the various biochemical and physiological ante mortem and post-mortem processes could be the best strategy for determining fish welfare during stunning/slaughtering procedures and their impact on product quality [24]. Hence, techniques for pre-slaughter and slaughter should be used to reduce the level of evoked stress response and physical activity [30].

### 2.2. Stunning Methods

The most important aspects of the slaughtering process are the handling prior to stunning and the immediacy of loss consciousness following stunning. Thus, stunning methods that cause immediate loss of consciousness and reduce exposure to aversive situations are considered fast methods, while the methods that do not cause immediate loss of consciousness are considered slow methods [31]. Therefore, due to the relationship between an animal’s welfare and subsequent meat quality, methods that cause a slow loss of sensibility have a negative impact on the carcass’ overall quality, whereas methods that cause a rapid loss of sensibility have a positive impact on the carcass’ overall quality [31].

Any killing protocol must be monitored to ensure its effectiveness, ensuring the least amount of fish suffering and allowing for the improvement of current methods, including the adoption of new techniques and improvements that take into account the quality of the flesh or operational costs [28]. In this sense, fish responses to stimuli and reflexes appear to be capable of distinguishing with reasonable certainty the state of awareness of a variety of species as they are killed [22,26]. Van de Vis et al. [32], on the other hand, believe that using behavioural indicators alone, even if they have been shown to correlate with brain activity, may not be sufficient, especially in ice slurry due to the muscle paralysis caused by cooling [31].

Thus, the goals of farmed fish slaughtering can be summarized as follows: reduce fear and pain, ensure product quality, and enable process efficiency [33]. Additionally, management during crowding prior to harvest should be conducted carefully, fish should be quickly unconscious, a large number of animals can be slaughtered in a short period of time without compromising welfare, and finally, consider the final destination of fish.

## 3. Criteria to Evaluate the Welfare Impact of Stunning Methods in Farmed Seabass and Seabream

### 3.1. Delay to Reach the Unconsciousness Stage

Slaughter methods, which do not cause immediate loss of consciousness, are primarily used in farmed seabass and seabream stunning/slaughtering. This is because they are simple and inexpensive to apply [31] and more appropriate in a wide range of fish. The most common process to render the fish less active and easier to manage during slaughter is the cooling of animals. Because the hypothermia procedure for stunning appears to be temperature-dependent [34], it may be more effective in the case of warm species such as seabass and seabream. Adding sedative agents, such as clove oil, to pre-slaughter cooling would go a step further in extending the effectiveness of fish stunning and reducing pain, though it is unclear whether this method would be acceptable to consumers, both in terms of sensory appeal and food safety. Some slaughter guidelines [35] accept the use of water plus ice slurry to stun seabass and seabream. The fish move around for a short time after being placed in the stunning tank before slowing down and becoming paralyzed as their muscles cool [31]. Despite the fact that the time required for sure stunning is not short, it does not appear to be particularly stressful in terms of causing pre-slaughtering cooling, reduced movements at death and gave good responses related to hematic and muscular stress indicators [36] with decreasing breathing amplitude and movement loss on the stunning tank’s bottom after 3 min [34]. Furthermore, fish are not asphyxiated in the ice slurry because they can breathe [31], and the paralysis caused by rapid cooling is reversible, with fish returning to rearing conditions quickly regaining muscular movement. In this sense, the stunning/slaughtering tank’s uniform water temperature of around 0 °C ensures that the fish do not die of asphyxia but rather of thermal shock [37].

The time required to detect the state of unconsciousness by using ice slurry to stun seabass varies between authors, ranging from 10 min [38] to 20 min [36,39]. The results for seabream are very similar, with time allocations ranging from 15 to 20 min [40]. Liquid ice or binary ice has been used to stun seabream [41,42,43] and seabass [44]. Fish are cooled faster than those stunned with ice slurry due to the physical properties of liquid ice (microscopic size of the ice crystals) and the low temperature of the stunning conditions. However, if the fish are kept in the ice slurry tank after stunning, the time to death would be around 25 [45] or 34 min [46], possibly reaching 40 min if the fish were crowded during pre-slaughter handling [44]. When compared to seabass, the time to death in seabream may be delayed by more than five minutes [44].

Another method used for seabass stun is the diffusion of gases into the stunning tank, faster to reach unconsciousness than ice slurry. In the case of carbon dioxide, the time it takes to reach unconsciousness will be 7 min, possibly as little as 4 min if the gas is insufflating into the ice slurry [36]. Aside from carbon dioxide, a mixture of nitrogen in various proportions has also been used but with no discernible differences [34]. Thus, the seabass dies after 16 [46] or 20 min [47] in CO_2_-supersaturated seawater, with the gas mixture reducing this time to 10 min [34].

The addition of anaesthetic has also been tested for stunning. The clove oil anaesthesia will render the fish unconscious before transferring them to an ice slurry for slaughter [38]. Recently, an innovative new option for reducing gilthead seabream stress at slaughter has been proposed: the inclusion of nanoencapsulated clove oil in the ice used to stunning in order to improve their water solubility [42].

Some electric stunning trials have also yielded promising results in seabream and sea-bass [31,34,36,48,49,50]. Following the application of an electric current, the fish are transferred to a tank filled with ice slurry for slaughter. In all cases, the fish are unconscious before being transferred to the slaughtering tank, though the recovery time varies depending on the methodology used, ranging from less than one minute [49] to more than twenty minutes [34]. This is especially important because a humane slaughter is achieved if the fish becomes insensible very quickly after being exposed to an electric field and remains insensible until death occurs [51].

The last three procedures for stunning seabream and seabass are asphyxia in air, spiking, and percussive stunning. Asphyxia should not be considered. It would not meet current animal welfare requirements for aquaculture [46] due to the lengthy time required for stunning (loss of movements only after more than an hour) and the violent reactions in the first minutes [36]. Spiking and percussive stunning, on the other hand, are not practical for batches of small-sized species, despite being the fastest and least stressful of all previously cited methodologies [36,52].

### 3.2. Metabolic Indicators of Stress

The stunning/slaughtering methods cause significant changes in plasma stress indicators, such as osmolality, glucose, lactate, and cortisol, reaching higher levels than the undisturbed fish in all cases [46]. However, depending on the method of stunning/slaughtering and the species, the magnitude of these changes varies. Even the stocking density and time spent in confinement prior to harvesting are factors to consider [53]. Thus, there were no differences in mean plasma cortisol, glucose, or lactate concentrations in seabream after electrical stunning followed by immersion in ice slurry or ice slurry alone [50]. Cortisol and lactate levels were higher in seabass stunned by electricity than in those immersed in ice slurry [54].

When it comes to using carbon dioxide to stun seabass, the results vary between authors, with some reporting higher levels of cortisol and lactate [36] and others reporting lower levels of lactate [46] when using the gas instead of ice slurry. This last finding is supported by a longer time to death, which could lead to increased metabolic activity in muscles and, as a result, the accumulation of metabolites such as plasma lactate. When the gas, nitrogen, or a mixture of nitrogen and carbon dioxide, was dissolved in the ice slurry, glucose and lactate did not differ significantly with or without gas, but cortisol was higher in the ice slurry alone, most likely due to the longest time for the fish to become stunning [39].

The use of anaesthetics in ice slurry produced different results in seabass and seabream. While plasma lactate levels in seabass stunned with clove oil were higher than in ice slurry [55,56], they were lower in seabream (Lopez-Cánovas et al., 2019). Furthermore, both plasma glucose and cortisol levels were lower after adding clove oil to seabream [42] or seabass [56], but only under experimental conditions, with no significant variations in serum cortisol levels detected under industrial farm conditions, most likely due to greater individual variability.

### 3.3. Flesh Quality

A way to evaluate the fish welfare during the slaughter is to analyse the impact on related aspects such as flesh quality parameters. The slaughter methods have a significant impact on the quality of the flesh, changing its physical properties, spoilage processes during ice storage, and sensory attributes [57]. Taking into account the impact of pre-slaughter stress on commercially harvested fish; however, the expected variations in quality parameters will be difficult to record and will most likely go unnoticed alongside the changes caused by slaughter [58]. As a result, various authors have investigated the effects of the stunning/slaughter method on the onset and resolution of rigor mortis, the evolution of post-mortem pH, freshness indicators such as the K-value (based on ATP breakdown and subsequent by-products) or the QIM (Quality Index Method) and, in some cases, other physical parameters, such as texture and colour.

In terms of rigor mortis, Knowles et al. [48] concluded that electrical stunning accelerated the pattern of onset and resolution of rigor mortis in seabass when compared to immersion in ice slurry, despite the fact that the time to stun is shorter [34]. The process was even faster when carbon dioxide was used [36]. This gradual onset of rigor mortis can provide information on the fish’s stress status prior to death while also preserving cellular energetic reserves [36]. Given the lower muscle activity and energy reserves consumption, the ice slurry would be appropriate for seabass [34]. In this regard, combining clove oil anaesthesia with immersion in ice slurry reduced anaerobic glycolytic activity and delayed the onset of rigor mortis [38], with a significant effect on ultimate muscle pH [58,59].

The evolution of pH is linked to rigor and energy consumption and is influenced by factors such as overcrowding and oxygen availability prior to slaughter [60]. When compared to ice slurry, seabass stunned by carbon dioxide showed a sharper decrease in muscle pH during the first hours [46]. Giuffrida et al. [40], on the other hand, found no significant differences in muscle pH values among seabream stunned/slaughtered with carbon dioxide or ice slurry. Electrical stunning seabass yielded contradictory results, with an initially lower pH compared to ice slurry or no differences between the two methods [48]. Percussive stunning [34] and asphyxia [39,61] were used to reach the extremes of highest and lowest pH values, respectively.

The K-value, which measures the concentration of muscle ATP and its degradation metabolites, can be used to detect slaughtering distress [24]. During storage of seabream using ice slurry, higher values of the ratio ATP/IMP are observed than when using carbon dioxide [40]. Sea bass stunned/slaughtered with ice slurry had higher IMP concentrations and lower levels of inosine and hypoxanthine than fish stunned/slaughtered with electricity, seawater, or flake ice saturated with a mixture of nitrogen and carbon dioxide, indicating a better freshness condition in ice slurry fish [34]. When compared to ice slurry, liquid ice maintains higher K-value values in seabream after slaughter [43]. Furthermore, the K-value of seabream killed by immersion in ice slurry or a blow to the head after anaesthesia with clove oil did not differ during chilling storage [62].

Depending on the authors, the results of the sensory evaluation of freshness, whether using the QIM methodology or similar protocols, were slightly different. While no differences were found between fish stunned/slaughtered on an ice slurry and fish stunned/slaughtered on carbon dioxide [46] or electricity [48], in some experiments with seabass, the fish stunned/slaughtered on an ice slurry maintained higher freshness scores during the shelf-life in others [34,36]. Seabass slaughtered in liquid ice showed more distinct differences, with significantly lower spoilage rates than fish slaughtered in ice slurry [44]. After percussive stunning, the QIM scores in seabream were similar to those obtained after immersion in an ice slurry (van de Vis et al., 2003).

Electrical stunning has been found to have a lower hardness than ice slurry in studies of seabass flesh texture variations [50]. The addition of clove oil to the rearing tank reduced pre-slaughter harvesting stress in seabream, but the hardness of the fillet, both raw and cooked, did not differ significantly from the stress condition caused by harvest net crowding [59]. Despite the fact that intense exercise prior to slaughter alters post-mortem muscle degradation processes via changes in myofibrillar proteins [63,64], the stunning/slaughtering method had no effect on skin lightness and colour in seabass [48,49] or seabream [43].

## 4. Conclusions

Consumers are becoming increasingly concerned about the treatment of the animals that provide us with food. Aquaculture’s higher efficiency in obtaining protein when compared to other farmed terrestrial animals has made it not only an important source of protein for human consumption but also a rapidly growing activity around the world. In this context, self-imposed good practice guidelines in aquaculture farms may be considered consistent with fish welfare outcomes, ensuring reproductive success, good growth performance, and product quality from a market standpoint. However, the most important aspect of animal welfare, the stunning/slaughter procedure, which is well-known and studied in other farmed terrestrial animals, is not well-studied in fish, especially seabass and seabream. In these species, some peculiarities arise in some stages of the process since, during the crowd and harvest procedures in the marine cages, the fish are transferred to the stunning tanks exhibiting a frenzied escape behaviour. Only the reduction of the time available in both processes helps to reduce the suffering of the fish.

The scientific evidence from the last two decades in these species supports stunning with ice slurry to promote hypothermia while paying attention to the flesh quality parameters as indicators of metabolic stress, which offer similar results to those associated with electrical stunning. Other methods that could be used on a commercial scale, such as asphyxia or carbon dioxide (alone or in combination with nitrogen), had more negative consequences. However, the most important question to consider is whether a slow slaughter method can be accepted due to the time it takes for the animal to become unconscious. The use of ice slurry in conjunction with a nanoencapsulated anaesthetic can help reduce fish stress during stunning and achieve an appropriate state of unconsciousness in seabass and seabream more quickly, all while remaining compliant with animal welfare standards.

## Data Availability

Not applicable.

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
