# Peer review of "Twenty Years of Research in Seabass and Seabream Welfare during Slaughter"

_animals, 2021, doi:10.3390/ani11082164_

Round 1

Reviewer 1 Report

This is a good topic for review, especially when focused more narrowly on sea bass and sea bream where the information can be brought together cohesively. There are some organizational issues that make it hard for the reader to follow the manuscript and fully grasp the impact of these practices on the welfare of the animals.

Title- the "/" is awkward and the title is therefore awkward. Consider re-phrasing "Twenty years of research in seabass and seabream welfare during slaughter" Or something similar.

Line  8- Simple Summary- The summary focuses on aspects of the pre-slaughter process that are only addressed briefly in the paper. The summary should better encompass all the aspects of the paper.

The abstract is disorganized and could do a better job describing how the manuscript first covers the ethical grounds for considering fish welfare at slaughter, then covers preslaughter welfare and then each common method of stunning, then a conclusion.

Introduction

At some point it would be warranted to explain how the post slaughter flesh quality parameters are related to the welfare of the fish. This may be in the introduction or elsewhere but it is currently not clear how meat quality parameters are related to welfare. If it is related more to whether these methods would be accepted by the industry, that should be explained.

Line 35- The question, what happens to fish after slaughter seems to be out of place here since it is not tied back to welfare (fish are dead after slaughter=no welfare) and if it is meant to tie the post slaughter flesh quality back to welfare parameters pre-slaughter, that is not clear. 

Line 54 to 59- A more nuanced discussion of whether fish have the anatomic structures capable of pain perception is warranted. The authors seem to argue that fish are not conscious of pain, which then calls the whole manuscript into question as it removes the moral grounds for concern about humane slaughter. A more detailed discussion of fish neuro-anatomy and their early evolutionary split from mammals (as discussed in Lund, 2007) would be warranted here.  Line 55 is particularly confusing and should be re-worded. A conclusion that fish, despite their different anatomy from mammals, still are likely conscious of painful stimuli, should be supported in order to move on to the application of the pre-cautionary principle and the grounds for the rest of the manuscript.

Line 63, "regardless of the foregoing" is confusing and unclear what it is referencing. 

Line 79- "thoughtfulness..." this is a long title for a section which describes the impact on welfare of various pre-slaughter handling and stunning techniques. Consider rephrasing.

Line 80- This should be where humane slaughter definition and parameters are introduced. Consider using the AVMA humane slaughter guidelines (or some other that introduces such parameters) and applying those criteria to fish. This citation is related more to the ethical goals whereas the authors are moving on in the manuscript to looking at the application of scientific parameters and their relationship to fish welfare.

Line 86-88 is confusing and not a complete sentence. 

Line 88-90 states that inducing unconsciousness can't cause suffering if it takes a long time. That is not a true statement. Maybe the authors intend to say that it "should not" be painful if it takes a prolonged period in order to be considered humane slaughter.

Line 96- Consider starting a new section on pre-slaughter welfare or feed withdrawal and handling. 

Line 100-112- Instead of talking about what methods should be used it would be more informational to the reader to hear about what methods have been used to examine pre-slaughter handling and stress in the species discussed in this paper and how that relates to their welfare and best practices that should be applied to these aspects of the slaughter process.

Line 113- According to welfare- This is an odd phrase. Consider changing. Also, consider only discussing pre-slaughter welfare here and discussing stunning and killing techniques in the next section.

Line 113-134 Consider starting a section on Stunning methods and their impact on welfare. Discussing the basis for whether a  method is humane seems repetitive with lines 80-90. Some stunning methods, such as ice slurry are going to limit the behavioral repertoire of the fish based on the fact that they cause paralysis. This renders behavior less useful as an assessment tool for such methods. This should be stated more clearly.

Line 117-119- Confusing. Consider tying the statement to a method instead of discussing "fast" vs "slow" methods.

Line 119 to 121 is lacking a citation and it is still unclear how this relates to welfare. Consider tying this statement to the methods under discussion.

Line 135- Consider changing the title of this section to cover welfare impact of various stunning methods. "convenience" is how easy something is for the industry and it is unclear how this relates to welfare.

Line 134- "slow methods" is imprecise. Explain which methods are being discussed. 

Line 138-148 Consider leaving discussion of anesthetic agents until later and using this paragraph to focus on the impact of temperature on the effectiveness of the stunning process.

Line 149-159- Stunning with ice "does not appear to be particularly stressful" is a strong statement. If this is based on behavioral observations then this should be stated, otherwise it should be more clear how we know it is not stressful. It is also unclear how the fact that it is reversible means that it is not stressful. Many other reviews would say that we do not have good evidence to say whether it is stressful or not.

Line 160-162- How is this assessment method proposed by Kestin related to the rest of the paragraph since the only metric mentioned is the time to unconsciousness. Were these methods applied in these other studies? Consider a paragraph for sea bass and one for sea bream as it is confusing which sentences apply to each type of fish.

Line 172-178 could be more clear whether addition of these various gases speeds up time to unconsciousness compared to ice alone.

Line 179- Remove "as a result"

Line 182- What were the effects on the fish of the clove oil and how was it improved by nano-encapsulation?

Line 184-191- What was the impact on the fish of electric stunning? did the methods in these studies vary with some being more effective? How does the duration of unconsciousness relate to the time it takes to get to the killing method? 

Section 3.2  and 3.3 - Consider whether having these in a separate section is better or whether they should be incorporated into  a section on each method. In my opinion, having it separated into methods of assessment makes it harder to come to a conclusion about the welfare impact of each method as its impacts on the fish are separated into multiple areas.

Section 3.3- It is unclear how each of these metrics relates to fish welfare.

Line 233 QIM needs to be defined. 

Conclusion- Overall, could do a better job summarizing the best practices for pre-slaughter welfare in these fish, the best stunning methods in these fish, and the impact that would have ethically and in terms of consumer perception as well as on the industry in terms of uptake of these practices based on the impact on flesh quality.

The section (line 291-295) on preslaughter welfare should reflect the best practices the authors feel are supported by the information they have reviewed.

Line 296-298 Confusing. Relates electrical stunning to a metabolic stress indicator but it is a stunning method. 

Author Response

To response to Reviewer 1 Comments, please see the attachment 

Reviewer 2 Report

MS is very interesting, well planned and written. I only miss a paragraph about the importance of sea bass and sea bream aquaculture (e.g. according to FAO data). Secondly, I miss a graphic or tabular presentation of the methods used, their advantages and disadvantages. It would greatly facilitate the reception of work.

My comments are attaching in the MS file as a commentaries.

Author Response

To response to Reviewer 2 comments, please see the attachment 
